# A Hierarchical Blockchain-Assisted Conditional Privacy-Preserving Authentication Scheme for Vehicular Ad Hoc Networks

**DOI:** 10.3390/s22062299

**Published:** 2022-03-16

**Authors:** Xingyu He, Xianhua Niu, Yangpeng Wang, Ling Xiong, Zhizhong Jiang, Cheng Gong

**Affiliations:** 1College of Computer and Software Engineering, Xihua University, Chengdu 610039, China; rurustef1212@gmail.com (X.N.); wangyangpeng089@gmail.com (Y.W.); lingxiong.swjtu@foxmail.com (L.X.); xhu_jzz@163.com (Z.J.); 15223789263@163.com (C.G.); 2National Key Laboratory of Science and Technology on Communications, University of Electronic Science and Technology of China, Chengdu 611731, China

**Keywords:** privacy-preserving, hierarchical, authentication, blockchain

## Abstract

Through information sharing, vehicles can know the surrounding road condition information timely in Vehicular Adhoc Networks. To ensure the validity of these messages and the security of vehicles, the message authentication, privacy-preserving, and delay problems are three important issues. Although many conditional privacy-preserving authentication schemes have been proposed to ensure secure communication, there still exist some imperfections such as frequent interactions or unlinkability. From this, our paper proposes a novel hierarchical blockchain-assisted authentication scheme to solve these existing issues comprehensively. First, unlinkability is achieved by a dynamic key derivation algorithm. Second, the proposed scheme can reduce correlation processing delay, queuing delay, and deployment costs by adopting hierarchical Vehicle Fog Computing. Third, cross-region authentication is achieved by taking advantage of the properties of blockchain. In addition, we demonstrate our scheme can fulfill the security criteria of the Vehicular Adhoc Network by security analysis. Furthermore, the simulations are carried out to show its availability by using JAVA and NS-3. The findings reveal that the suggested method outperforms earlier schemes in terms of computation cost and communication cost. All in all, making the authentication scheme more efficient and concise is the focus of our future research.

## 1. Introduction

Vehicular Adhoc Networks(VANETs) were proposed to ease traffic pressure and reduce traffic accidents [1,2]. VANETs take vehicles as the basic information units and realize the network connection between vehicles and X (e.g., vehicles, infrastructures) through the help of the new generation of information and communication technology. Figure 1 is a typical VANET. There are two basic modalities of communications, namely: Vehicle-to-Vehicle (V2V) and Vehicle-to-Infrastructures (V2I). By sharing information about the surrounding road conditions, other vehicles can replan their routes in time to avoid traffic jams and traffic accidents after receiving these messages. In addition, the Traffic Control Center (TCC) can make flexible adjustments to traffic timely to ease the traffic pressure. VANETs are seen as a potential technology in current intelligent transportation systems because of these benefits.

Since communications are exchanged wirelessly via open channels, it is simple for an attacker to intercept messages from communication channels and launch a series of harmful assaults (e.g., impersonate a legitimate vehicle to send a false message or tamper with messages) [3,4,5,6]. Once the recipient makes a wrong decision based on these malicious messages, it may lead to traffic jams or even car accidents. In addition, the identity information of vehicles, itinerary, and other factors may be used by the adversary to carry out an attack. Hence, security and privacy are two important factors that cannot be ignored in VANETs. In other words, the authenticity and the legitimacy of the messages should be guaranteed in VANETs.

To address the privacy and security issues that plague VANETs, a significant variety of privacy-preserving authentication schemes have been presented during the previous decade [7,8,9,10]. These solutions go a long way toward overcoming issues like conditional privacy and unlinkability. At the same time, it also provides the direction and theoretical basis for the subsequent researchers. Although these solutions can solve the problem of cross-region authentication, additional communication is required during cross-region authentication, which increases the related cost.

Blockchain technology provides a good idea for cross-region authentication. Lin et al. [11] adopted a dynamic key derivative algorithm to realize the unlinkability. Compared with the traditional scheme, this scheme can not only realize fast cross-region authentication but also avoid storing a large number of pseudonyms. However, in order to speed up the process, their scheme requires all RSUs to be trusted, which may increase the deployment costs of the infrastructure. In addition, this scheme has the defects of high queuing delay in theory due to all communications of vehicles being processed by a single trusted authority (TA).

With the increasing number of connected vehicles, Vehicle Fog Computing (VFC) was proposed to satisfy low latency and uninterrupted services for users. In this pattern, data, processing, and applications are centralized in devices at the edge of the network and can significantly reduce the delay of message processing. Furthermore, that is why VFC is considered as a promising technology to make the best utilization of these vehicular communication and computational resources [12,13]. Yao et al. [14] and Kaur et al. [15] adopted distributed VFC to reduce the delay by using multiple regional managers to handle vehicles’ communications instead of a single TA. Nonetheless, they did not take into account the unlinkability of messages. As a result, to assure the safety and privacy of vehicles and to reduce delay to realize real-time communications in VANETs, we designed a hierarchical effective blockchain-assisted conditional privacy-preserving scheme. The proposed scheme can reduce the time delay and deployment costs by using hierarchical distributed VFC. Furthermore, unlinkability is achieved through a dynamic key derived algorithm.

### 1.1. Contribution

In this study, we proposed a hierarchical blockchain-assi-sted conditional privacy-preserving authentication (CPPA) scheme for VANETs. The following are the paper’s significant contributions:First, to reduce the deployment costs and the associated processing time, the proposed scheme adopted distributed VFC, which uses multiple regional managers instead of a single TA.Second, to achieve unlinkability, the proposed scheme employs a dynamic key derivation algorithm to generate dynamic public-private key pairs for each communication of vehicles.Third, through Java and NS-3 simulation experiments, we show that our scheme is suitable for VANETs in terms of communication and computing overhead.

### 1.2. Organization

The following is a rough outline of the paper’s structure. Section 2 examines relevant CPPA schemes for VANETs. Section 3 presents the relevant preliminary knowledge for our scheme. Our system model and security and privacy requirements for VANETs and the details of our scheme are shown in Section 4. Security analysis is carried out in Section 5. In Section 6, we examine the corresponding computing and communication overhead and compare it to existing schemes. In the end, we provided a brief summary of this study in Section 7.

## 2. Related Work

To achieve effective communication in VANETs, many authentication schemes have been proposed. Picconi et al. [16] relied on PKI-based authentication, proposed a solution for validating aggregated data in V2V traffic information systems. Zhang et al. [17] adopted the *k*-anonymity to protect user identity privacy, while they are useful to some extent in addressing privacy issues VANET, the difficulties of certificate management make them impractical. To improve the efficiency of ID-based CPPA scheme, He et al. [8] proposed a novel CPPA scheme by utilizing the elliptic curve cryptography(ECC). Zhong et al. [18] employs pseudonym-based signatures for identity authentication for VANET. Furthermore, both of them also support batch validation, allowing the verifier to validate multiple messages simultaneously, which greatly improves the validation efficiency. However, then, these programs require a pre-established trust relationship between the regional management center and vehicles, which will not exist once a vehicle moves to another region. Wang et al. [19] adopted group-based message authentication algorithm to address the security issues in V2V communication. When a vehicle enters the coverage area of a new RSU, it must be re-authenticated to the new RSU, which undoubtedly increases the delay. Ali et al. [20] design an efficient conditional privacy-preserving hybrid signcryption scheme for heterogeneous vehicle communication based on bilinear pairing. The users’ privacy is protected to a certain extent, but the unlinkability cannot be guaranteed.

With the popularity of blockchain technology, many scholars employ blockchain to realize cross-region authentication. Yao et al. [14] proposed a BLA for distributed VFS to achieve a flexible cross-region authentication. The public key and identity information of vehicles are placed on a consortium blockchain so that different regional managers can verify the messages sent by legal vehicles and then provide VFS for them. Kaur et al. [15] present an effective cross-region authentication and key-exchange scheme based on Yao [14], which realize mutual authentication between vehicles and service managers. However, unlinkability cannot be satisfied. Wang et al. [21] present trustworthiness evaluation to achieve a time-efficient V2I-handover authentication. It seems only considered V2I communication scenarios. Lu et al. [22] adopted the Merkle Patricia tree (MPT) to extend the conventional blockchain and record the activities of the semiTAs in blockchain to achieve the certificate and revocation transparency. However, it requires vehicles to interact frequently with the certificate center to generate anonymous certificates, resulting in low efficiency. Inspired by the HD Wallet, Lin et al. [11] proposed a novel BCPPA by using key derivation algorithms and smart contracts. The public key certificate of each communication of vehicles is pre-placed in the blockchain for vehicles to retrieve, saving the overhead of storing a large number of certificates in OBUs. However, it seems to be burdensome for Certificate Authorities (CA) to generate a public key certificate for every communication of all vehicles. For clarification, a brief summary is given in Table 1.

## 3. Preliminaries

The relevant preliminary knowledge was briefly present-ed in this part.

### 3.1. Broadcast Encryption

Broadcast encryption may be thought of as a type of key encapsulation scheme. In our proposed scheme, we adopt a broadcast encryption case from [23] to complete the identification of vehicles by legitimate SMs. The whole process can be divided into the following three parts.

(1)Setup: In our proposal, the maximum number of SMs is set to be *n*. Let *L* stand for a set where L⊆{1,⋯,n}. This step is mainly TA distributes a key $d_{j}$ for each $SM_{j}$, for j∈L. Then TA publish a public parameter PK.(2)Enc(PK,L): A vehicle want to jion the internet of vehicles, it uses PK to calculate the encryption key *K* and the header Hdr. The vehicle then encrypts a message *M* using *K* as a symmetric encryption key. Let $E_{k}\left(M\right)$ be the encryption of *M*. Finally, it broadcast $<E_{K}\left(M\right),Hdr>$.(3)Dec($PK,L,k,d_{k},Hdr$): Let $SM_{k}$ be an example to decrypt $E_{K}\left(M\right)$. If k∈L, by inputting PK, the set *L*, the key $d_{k}$ and Hdr, $SM_{k}$ can easily compute a message encryption key $K_{k}^{\prime }$. It’s remarkable that $K_{k}^{\prime }=K$, this indicates that $SM_{k}$ can decrypt $E_{K}\left(M\right)$ and retrieve M by using $K_{k}^{\prime }$.

### 3.2. Mathematical Complexity Assumptions

The security of broadcast encryption relies on the bilinear Diffie-Hellman Exponent Assumption (BDHE). Generally, we define ζ-BDHE in group $G_{1}$ as follows: input a vector with 2ζ+1 elements where $(h,g,g^{\alpha },g^{\left(\alpha^{2}\right)},\cdots ,g^{\alpha^{\zeta }},$$g^{\left(\alpha^{\zeta +2}\right)},\cdots ,g^{\left(\alpha^{2\zeta }\right)})\in G_{1}^{2\zeta +1}$. Then, output $e{\left(g,h\right)}^{\left(\alpha^{\zeta +1}\right)}\in G_{T}$. Since $g^{\left(\alpha^{\zeta +1}\right)}$ is not included in the vector, we can not able to compute the required $e{\left(g,h\right)}^{\left(\alpha^{\zeta +1}\right)}$ by the bilinear map.

For convenience, we let $g_{i}=g^{\left(\alpha^{i}\right)}\in G_{1}$ while *g* and α are given. An algorithm 𝒜 has ϵ advantage to tackle ζ-BDHE in $G_{1}$ if Pr[𝒜$(h,g,g_{1},\cdots ,g_{\zeta },g_{\zeta +2},\cdots ,g_{2\zeta })=$
$e(g_{\zeta +1},h)$] ≥ϵ. Note that *g* and *h* are random picks in $G_{1}$, α is random pick in $Z_{\bar{q}}$, and 𝒜 picks the bits randomly.

Homoplastically, the definition of Decisional ζ-BDHE(D-ζ-BDHE) in $G_{1}$ is as follows. Let $y_{g,\alpha ,\zeta }=(g_{1},\cdots ,$$g_{\zeta },g_{\zeta +2}$$,\cdots ,g_{2\zeta })$. An algorithm ℬ has advantage ϵ to output a bit ψ∈{0,1} to tackle D-ζ-BDHE in $G_{1}$ if |Pr$[B(g,h,y_{g,\alpha ,\zeta },$$e(g_{\zeta +1},h))=0]-$Pr$[B\left(g,h,y_{g,\alpha ,\zeta },T\right)=0]|\geq \epsilon$. Similarly, *g* and *h* are random picks in $G_{1}$, α is a random pick in $Z_{\bar{q}}$, *T* is a random pick in $G_{T}$ and B picks the bits randomly. Here, we let $P_{BDHE}$ to present the left and $R_{BDHE}$ to present the right.

### 3.3. Key Derivation

We anticipate that a vehicle will be able to employ distinct public and private key pairs to achieve unlinkability and not need to exchange keys or preload abundant key pairs. Therefore, we adopt a key derivation algorithm which proposed by [11]. This algorithm is separated into two parts: public key generation algorithm and private key generation algorithm. We have developed a flow chart in Figure 2 and included a quick explanation below to make it simpler to understand.

Private key generation algorithm: The goal of this algorithm is for vehicles to generate a private key for subsequent communication. A vehicle randomly selects a seed to create the root private key ($sk_{root}$) and root chain code ($chain_{root}$). Then calculates the appropriate root public key ($pk_{root}=sk_{root}\cdot P$) and sends <$pk_{root},chain_{root}$> to public key generator (i.e., SM). Based on $sk_{root}$, $chain_{root}$ and serial number of the current communication (*i*), the vehicle can derive a different private key.Public key generation algorithm: The purpose of this algorithm is for SMs to generate corresponding public keys for subsequent communication of vehicles. According to Figure 2, for the same serial number *i*, $pk_{i}=sk_{i}\cdot P$. It ensures that the public key retrieved by the verifier corresponds to the private key of the vehicle.

## 4. Scheme Description

In this section, we first introduced the system model and related security requirements, and then described our solution in detail. On the whole, The proposed scheme can mainly be divided into five phases, namely Initialization Phase, Registration Phase, Identity Authentication Phase, Consensus Phase and Message Authentication Phase. Note that we assume that there are *n* SMs in the system. Table 2 shows the definitions of the notations used in this article.

### 4.1. System Model

Figure 3 depicts our system model. In conclusion, we separated the entire system into a number of regions, and each region is managed by a single SM. The functions of each entity in our system are described as follows.

Trusted Authority (TA): TA is a completely trusted department that generally has strong computing and communication capabilities. In our system, TA is required to complete registration SMs and vehicles. If necessary, TA can find out the real identity of a malicious vehicle through a relevant message.Service Manager (SM): an SM is mainly responsible for the identification of new vehicles joining VANETs in its region. Furthermore, SM is also responsible for the calculation of public keys and pseudonyms for subsequent communication of its certified vehicles.Road Side Unit (RSU): RSU is a semi-trusted roadside infrastructure that can communicate wirelessly with vehicles according to the Dedicated Short Range Communication (DSRC) protocol [24]. Furthermore, RSUs are also responsible for forwarding messages between vehicles and SMs and providing VFS to vehicles.Vehicle: as moving nodes, vehicles are outfitted with On-Board Units (OBUs), which are wireless communication devices. OBU is a tamper-proof device that also has certain computation and communication capabilities. By using OBUs, vehicles may exchange their current road traffic circumstances and driving status with the adjacent vehicles and RSUs in real-time via DSRC protocol. What’more, OBU’s information will never be revealed.

### 4.2. Security and Privacy Requirements

Identity authentication: SM can effectively verify the legitimacy of new vehicles joining VANETs.Message authentication: for any received message, the verifier can verify that the message is valid.Identity privacy preservation: except for TA, no one can deduce the true identity of vehicles from the intercepted messages.Unlinkability: it will be impossible for a adversary to link two messages transmitted by the same vehicle.Traceability: if required, TA can determine the message sender’s true identity. This guarantees that messages are held accountable.Resist various attacks: our scheme also assures that oth-er assaults in VANETs, such as the replay attack, the impersonation attack and the modification attack, can be easily identified.

### 4.3. Initialization Phase

This phase is mainly performed by TA to creates a series of system parameters. The following are the specifics.

Picks two random large prime integers p,q and choose an additive group G generated by a point *P* with order *q* on a non-singular elliptic curve $E:y^{2}=x^{3}+vx+w$ mod *p* where $v,w\in F_{p}$.Picks two large prime numbers $\bar{p},\bar{q}$ at random and chooses two multiplicative groups $G_{1}$,$G_{T}$ generated by a point *g* with order $\bar{q}$.Selects a number $sk_{TA}\in Z_{q}^{*}$ at random as its private key, then computes $pk_{TA}=sk_{TA}\cdot P$ as its public key.Chooses a random number $\alpha \in Z_{\bar{q}}^{*}$, calculates $g_{\gamma }=g^{\alpha^{\gamma }}\in G_{1}$ for γ={1,2,⋯,n,n+2,⋯,2n}.Picks a random number $\beta \in Z_{\bar{q}}^{*}$, and then computes $v=g^{\beta }\in G_{1}$. Next, TA set $PK=(g,g_{1},g_{2},\cdots ,g_{n},$$g_{n+2},$$\cdots ,g_{2n},v)\in G_{1}^{\left(2n+1\right)}$.Choose a secure hash function *H*, where $H:G\to Z_{q}$.Finally, the TA sends the public parameters $\{G,G_{1},$$G_{T},q,P,g,pk_{TA},PK,H\}$ to all SMs and vehicles.

### 4.4. Registration Phase

This phase is mainly divided into SMs and vehicles registration. Figure 4 briefly depicts this process. It’s worth noting that this process only needs to be done once. An SM’s registration details are as follows.

Assume that *j*th SM’s real identity is $ID_{SM_{j}}$. It choos-es a random integer $sk_{SM_{j}}$ as its private key and calculates ${pk}_{SM_{j}}=sk_{SM_{j}}\cdot P$ as public key. Then sends <$ID_{SM_{j}}$, ${pk}_{SM_{j}}$> to TA through a secure channel.TA first checks the availability of $ID_{SM_{j}}$ in its databa-se after getting this registration request. If a match is found, TA will rejects this registration request. Otherwise, TA computes $d_{j}=v^{\alpha^{j}}$ for $SM_{j}$ to decrypt broadcast messages. Then TA stores $ID_{SM_{j}}$ and $pk_{SM_{j}}$ into its database and returns $d_{j}$ to $SM_{j}$ through a secure channel.$SM_{j}$ stores $d_{j}$ into its database.

The specific operations of a vehicle’s registration are as follows.

Assume that *i*th vehicle’s real identity is $ID_{V_{i}}$, $V_{i}$ first randomly chooses a private seed to generate the private information $\left(sk_{root}\;and\;chain_{root}\right)$. Then $V_{i}$ computes $pk_{root}=sk_{root}\cdot P$. Finally, $V_{i}$ sends $<ID_{V_{i}},$$pk_{root},chain_{root}>$ to TA via a secure channel.When the TA receives this registration request, it first checks whether $ID_{V_{i}}$ is vaild. If not, the TA will rejects this request. Otherwise, the TA issue a password PWD and a certificate *S* for $pk_{root}$ and $chain_{root}$, where $S=Sign_{sk_{TA}}(pk_{root}||chain_{root})$. Finally, the TA stores $<ID_{V_{i}},pk_{root},chain_{root}>$ into its repository and returns <S,PWD> to $V_{i}$ via a secure channel.$V_{i}$ save $<S,chain_{root},sk_{root}>$ into its repository.

### 4.5. Identity Authentication Phase

The main purpose of this phase is to authenticate the identity of vehices through the corresponding SMs. After verification, SMs will generates corresponding pseudonym and public key pairs for future communications of vehicles. The detailed process is shown in Figure 5. When a registered vehicle $V_{a}$ first access VFS, it will executes the following operations to complete the authentication.

Picks a random integer $r\in Z_{\bar{q}}^{*}$ and then computes symmetric encryption (e.g., AES) key $K=e{\left(g_{n},g_{1}\right)}^{r}$. Then $V_{a}$ sets the header $Hdr=\left(C_{0},C_{1}\right)\in G_{1}^{2}$, where $C_{0}=g^{r}$ and $C_{1}={\left(v\cdot \prod_{\mu \in L}g_{n+1-\mu }\right)}^{r}$.Calculate the signature $\vartheta =Sign_{sk_{root}}(ID_{V_{a}}\left|\right|tst$$||pk_{root}||chain_{root})$. where tst is current timestamp and $ID_{V_{a}}$ is real identity of $V_{a}$.Compute the ciphertext $CT=E_{K}(\vartheta ||ID_{V_{a}}\left||tst||S|\right|$$pk_{root}||chain_{root})$. Then $V_{a}$ sends <CT,Hdr,tst> to the nearest RSU, let us assume it is $RSU_{m}$ and its region manager is $SM_{k}$.$RSU_{m}$ will transmits <CT,Hdr,tst> to $SM_{k}$.

$SM_{k}$ will executes the following operations after receiving <CT,Hdr,tst> transmitted by $RSU_{m}$.

Check whether tst is fresh. $SM_{k}$ will reject this message if tst is not fresh.Compute $K_{k}^{\prime }=\frac{e(g_{k},C_{1})}{e(d_{k}\cdot \prod_{\omega \in L,\omega \neq k}g_{n+1-\omega +k},C_{0})}$. Then $SM_{k}$ executes $D_{K_{k}^{\prime }}\left(CT\right)$ to extract ϑ, *S*, $ID_{V_{a}}$, $pk_{root}$ and $chain_{root}$.Verify ϑ by using $pk_{root}$ and verify *S* by using TA’s public key. $SM_{k}$ will reject this message As long as there is a validation failure. Then $SM_{k}$ calculates corresponding pseudonyms and public keys pairs (${PID}_{V_{a}}$, ${pk}_{V_{a}}$) for $V_{a}$ future communications by excuting public key generation algorithm described in IV-D. Here $\left({PID}_{V_{a}},{pk}_{V_{a}}\right)$=$\left\{\left(PID_{V_{a_{1}}},pk_{V_{a_{1}}}\right),\left(PID_{V_{a_{2}}},pk_{V_{a_{2}}}\right),\cdots ,\left(PID_{V_{a_{z}}},pk_{V_{a_{z}}}\right)\right\}$, where *z* is the number of elements in each set. For any u∈{1,⋯,z}, $PID_{V_{a_{u}}}=\left(PID_{V_{a_{u,1}}},PID_{V_{a_{u,2}}}\right)$ where $PID_{V_{a_{u,1}}}=chain_{V_{a_{u}}}\cdot P$ and $PID_{V_{a_{u,2}}}=ID_{V_{a}}⊕H\left(chain_{V_{a_{u}}}\cdot pk_{TA}\right)$.Send $\left({PID}_{V_{a}},{pk}_{V_{a}}\right)$ to blockchain.

### 4.6. Consensus Phase

We assume that there are *n* SMs in our system, and all of them are trusted. Under the circumstances, SMs acts as commit nodes according to the serial number. As we discussed earlier, SMs will send these related messages to the blockchain each time it completes the derivation of the pseudonyms and public key pairs of the vehicle it certifies. Assume that the time to produce a block is τ. During this time, SMs briefly store these pseudonyms and public key pairs it receives in their own memory. After τ time, the current commit node will publish the relevant information it has stored in memory as a new full block. Any SM, upon receiving a block, deletes the information in its own memory that is duplicated with the block and then performs the next consensus.

### 4.7. Message Authentication Phase

At this phase, as long as $V_{a}$ communication number does not exceed Z, re-authentication is not required regardless of whether $V_{a}$ has left $SM_{k}$’s jurisdiction. Figure 6 gives a brief description of this process, and the details are described below.

Assume the current serial number of $V_{a}$ is *b*, $V_{a}$ first initiates a request to the OBU by entering PWD and $ID_{V_{a}}$. OBU will reject the request if it does not match its own stored information, otherwise it goes to the next step.OBU calculates the current private key private key $sk_{V_{a,b}}$ and $chain_{V_{a_{b}}}$ based on private key generation algorithm. Then, calculates the current pseudonym $PID_{V_{a_{b}}}=\left(PID_{V_{a_{b,1}}},PID_{V_{a_{b,2}}}\right)$ where $PID_{V_{a_{b,1}}}=chain_{V_{a_{b}}}\cdot P$ and $PID_{V_{a_{b,2}}}=ID_{V_{a}}⊕H\left(chain_{V_{a_{b}}}\cdot \;PK_{TA}\right)$.Finally, returns $<sk_{V_{a,b}},PID_{V_{a_{b}}}>$ to $V_{a}$.Upon receiving the above information, $V_{b}$ calculates signature $\delta =\{Sign_{sk_{V_{a_{b}}}}\left(M||tst|\right|$$PID_{V_{a_{b}}})$ and then broadcasts the message $\left\{\delta ,M,tst,PID_{V_{a_{b}}}\right\}$.When a receiver wants to verify the message, it first checks whether the timestamp is valid. If valid, search for the corresponding public key $pk_{V_{a_{b}}}$ on blockchain according to the pseudonym. The receiver will rejects the message if the query fails. Finally, the receiver validates the signature by using $pk_{V_{a_{b}}}$. If the authentication succeeds, the message is trusted.

## 5. Security Analysis

We examine the security of our proposed VANET system in light of the design goals set out in Section 3. The details are given as follows.

### 5.1. Correctness

For the proof of the correctness of our proposed scheme, we need to verify $K_{k}^{\prime }=k$ to ensure $SM_{k}$ can decrypt the message send by $V_{a}$.The details are as follows.
(1)$$\begin{array}{cc} K_{k}^{\prime }= & \frac{e(g_{k},C_{1})}{e(d_{k}\cdot \prod_{\omega \in L,\omega \neq k}g_{n+1-\omega +k},C_{0})}\\ = & \frac{{\left(g,g\right)}^{\alpha^{k}\cdot r\left(\beta +\sum_{k\in L}\alpha^{n+1-k}\right)}}{e{\left(g,g\right)}^{r\cdot (\beta \alpha^{k}+\sum_{\omega \in L,\omega \neq k}\alpha^{n+1-\omega +k})}}\\ = & \frac{e{\left(g,g\right)}^{r\cdot (\beta \alpha^{k}+\sum_{k\in L}\alpha^{n+1-k})}}{e{\left(g,g\right)}^{r\cdot (\beta \alpha^{k}+\sum_{\omega \in L,\omega \neq k}\alpha^{n+1-\omega +k})}}\\ = & e{\left(g,g\right)}^{r\alpha^{n+1}}\\ = & K \end{array}$$

### 5.2. Security Model

We define chosen ciphertext security (CCS) of a broadcast encryption system against a static adversary. Security is defined by following game between an algorithm 𝒜 and a challenger. In addition, *n* (i.e., the total number of users.) is the input for algorithm 𝒜 and the challenger C.

**Init.** 𝒜 generates a set $L^{*}\subseteq \left\{1,\cdots ,n\right\}$ of users that it wishes to assault.

**Setup.**C executes Setup(n) to gain PK and $d_{1},\cdots ,d_{n}$. Then, sends PK and all $d_{f}$ to 𝒜, where $f\notin L^{*}$.

**Query phase 1.** 𝒜 sends out adaptive decryption queries $q_{1},\cdots ,q_{m}$. Here, (u,L,Hdr) is included in all decryption queries where $L\subseteq L^{*}$ and u∈L. Then, C returns Dec(PK,

$L,u,d_{u},Hdr)$ as response.

**Challenge.**C executes Enc(PK,L) to generate ($Hdr^{\prime },$
*K*) where $Hdr^{\prime }$ is another header, K is a finite key set and K∈K. Then, selects a bit ψ∈{0,1} randomly. Next, sets $K_{\psi }=K$ and selects a random $K_{1-\psi }\in K$. Finally, returns ($Hdr^{\prime },K_{0},K_{1}$) to 𝒜.

**Query phase 2.** Adaptively, 𝒜 send out more decryption queries $q_{m+1},\cdots ,q_{q_{D}}$ where $q_{i}=\left(u,L,Hdr\right)$ for $L\subseteq L^{*}$ and u∈S. Notice that $Hdr\neq Hdr^{\prime }$. Then, the C returns same response as phase 1.

**Guess.** 𝒜 outputs its $\psi^{\prime }\in \left\{0,1\right\}$ guess regarding ψ. If $\psi =\psi^{\prime }$ is held, 𝒜 will win this game.

Here, let **AdvBr**_*𝒜,n*_ represent the probability of 𝒜 wins the game.

**Definition** **1.**
*The broadcast encryption is $\left(t,\epsilon ,n,q_{D}\right)$ chosen ciphertext attack(CCA) secure if for all t-time algorithm 𝒜 make $q_{D}$ times decryption queries, we have that |*
**
*AdvBr*
**
_
*𝒜,n*
_
*$-\frac{1}{2}$|<ϵ.*


Similarly, we define semantic security of the broadcast encryption by preventing the attacker from issuing decryption queries.

**Definition** **2.**
*The broadcast encryption system is (t,ϵ,n) semantically secure assuming it is (t,ϵ,n,0) CCA secure.*


**Definition** **3.**
*The D-(t,ϵ,ζ)-BDHE assumption is hold in $G_{1}$ assuming no t-time algorithm has at least ϵ advantage to tackle the D-(t,ϵ,ζ)-BDHE problem in $G_{1}$.*


### 5.3. Formal Analysis

**Theorem** **1.**
*For any postive integers I, n (n>I), our I-broadcast encryption system is (t, ϵ, n) semantically secure if the D-(t, ϵ, I)-BDHE assumption is hold in $G_{1}$.*


**Proof of Theorem 1.** Assuming 𝒜 is a *t*-time adversary, for a given *I*, **AdvBr**_*𝒜,n*_>ϵ is hold. Build a new algorithm B that has ϵ advantage to tackle the D-*I*-BDHE problem in $G_{1}$. B picks a random D-*I*-BDHE challenge (*g*, *h*, $y_{g,\alpha ,I}$, *Z*) as input, where $y_{g,\alpha ,I}$=($g_{1}$,..., $g_{I}$, $g_{I+2}$,..., $g_{2I}$) and *Z* is either $e(g_{I+1},h)$ or a random member of $G_{T}$. The following is how B continues.**Init.**B executes 𝒜 and get L of users that 𝒜 wants to assault.**Setup.**B is responsible for generating PK and all $d_{i}$ for i∉L. Let $j=\lceil \frac{n}{I}\rceil$. Note that the choice of $v_{1},\cdots ,v_{j}$ is the key of the proof.B picks random $u_{i}\in Z_{\bar{q}}$ for 1≤i≤j. Here, we define two subsets $\hat{L_{i}}$ and $L_{i}$ as $\hat{L_{i}}=L\cap \left\{\left(i-1\right)I+1,\cdots ,iI\right\}$ and $L_{i}=\left\{x-iI+I|x\in \hat{L_{i}}\right\}\subseteq \left\{1,\cdots ,I\right\}$, respectively. For 1≤i≤j, B makes $v_{i}=g^{u_{i}}{\left(\prod_{f\in L_{i}}g_{I+1-f}\right)}^{-1}$. Then, B returns the public key PK to 𝒜, where $PK=(g,g_{1},\cdots ,g_{I},\cdots ,$$g_{2I},v_{1},\cdots ,v_{j})\in G_{1}^{2I-j}$. For all i∉L, we set i=(a−1)I+b for some 1≤a≤j and 1≤b≤I. B calculates $d_{i}=g_{b}^{u_{i}}\cdot \prod_{f\in L_{a}}{\left(g_{I+1-f+b}\right)}^{-1}$. Here, we get $d_{i}={\left(g_{b}^{u_{i}}\cdot \prod_{f\in L_{a}}{\left(g_{I+1-f}\right)}^{-1}\right)}^{\left(\alpha^{b}\right)}=v_{a}^{\left(\alpha^{b}\right)}$ as required. Furthermore, we know that $b\notin L_{a}$ because of i∉L, so $d_{i}$ does not include the term $g_{I+1}$.**Challenge.**B sets Hdr as $\left(h,h^{\left(u_{1}\right)},\cdots ,h^{\left(u_{j}\right)}\right)$. Then, cho-oses a random bit ψ∈{0,1} and makes $K_{\psi }=Z$ and randomly selects a $K_{1-\psi }\in G_{T}$. Finally, returns ($Hdr,K_{0},K_{1}$) as the challenge to 𝒜. We claim (Hdr,$K_{0}$,$K_{1}$) is a reasonable challenge to 𝒜 while $Z=e(g_{I+1,h})$ (i.e., the input to B is a B-BDHE tuple from $P_{BDHE}$ sampling). Moreover, for some (unknown) $t\in Z_{q}$, we write $h=g^{t}$. Then, for all i=1,⋯,j, we have $h^{u_{i}}={\left(g^{u_{i}}\right)}^{t}={\left\{g^{u_{i}}\cdot {\left(\prod_{f\in L_{i}}g_{I+1-f}\right)}^{-1}\cdot \left(\prod_{f\in L_{i}}g_{I+1-f}\right)\right\}}^{t}={\left(v_{i}\cdot \prod_{f\in L_{i}}g_{I+1-f}\right)}^{t}$. Here, for key $e{\left(g_{I+1},g\right)}^{t}$, ($h,h^{u_{1}},\cdots ,h^{u_{j}}$) is a vaild encryption. Furthermore, since $e{\left(g_{I+1},g\right)}^{t}=e(g_{I+1},$$h)=Z=K_{\psi })$, (Hdr,$K_{0}$,$K_{1}$) is a reasonable challenge to 𝒜. In addition, $K_{0},K_{1}$ are both picked randomly from $G_{T}$ while *Z* is chosen randomly from $G_{1}$ (i.e., the input to B is a B-BDHE tuple from $P_{BDHE}$ sampling).**Guess.** 𝒜 outputs $\psi^{\prime }$ guess regarding ψ. B outputs 0 if $\psi^{\prime }=\psi$ (i.e., $Z=e(g_{I+1},h)$). Otherwise, outputs 1 (i.e., *Z* is a random element in $G_{T}$). We can find that Pr$[B\left(g,h,y_{g,\alpha ,I},Z\right)$$=0]=\frac{1}{2}$ if $\left(g,h,y_{g,\alpha ,I},Z\right)$ is sampled from $R_{BDHE}$. Similarly, |Pr$\left[B\left(g,h,y_{g,\alpha ,I},Z\right)=0\right]-\frac{1}{2}|=$${\mathrm{AdvBr}}_{A,n}\geq \epsilon$ if $\left(g,h,y_{g,\alpha ,I},Z\right)$ comes from $P_{BDHE}$. Therefore, B has advantage at least ϵ to tackle D-*I*-BDHE in $G_{1}$. So we have proved the Theorem 1. □

### 5.4. Nonformal Analysis

**Identity authentication:** Because of the security of ECDSA, without TA’s private key, no one can impersonate TA and fabricate a certificate. Therefore, SMs can determine whether the vehicle is a legitimate vehicle as long as it verify the certificate sent by the vehicle with TA’s public key.**Message authentication:** According to the security of ECDSA, without signature key, no one can fake a legitimate message. It simply implies that a verifier only needs to compare the received signature message to the appropriate key to verify whether the message is legitimate.**Identity privacy preservation:** In **Identity Authentication phase**, Only legal SMs are able to decode the communication and determine a vehicle’s true identification. In **Message Authentication phase**, this real identity will be hidden under a pseudonym, which means no one(besides TA, who has its private key $sk_{TA}$) could deduce a vehicle’s true identification from the delivered communications. Therefore, our scheme can satisfiy the identity privacy preservation.**Unlinkability:** The vehicle produces a new private key and pseudonym each time it transmits a message *M*, then signs M. Because the pseudonym is linked to the chain code of vehicles, to link two message delivered by the same vehicle, adversary should own the chain code $chain_{root}$ and $pk_{root}$ of this vehicle. However, this two information only known by TA and the regional managers. Hence, our proposal can meet the security property requirement of Unlinkability.**Traceability:** Note that only TA owns its private key $sk_{TA}$. Since $sk_{TA}\cdot ID_{V_{a_{b,1}}}=sk_{TA}\cdot chain_{V_{a_{b}}}\cdot P=chain_{V_{a_{b}}}\cdot pk_{TA}$, when necessary(e.g., a vehicle sends a error messages that causes a traffic accident), TA can retrieve the vehicle’s genuine identifying information. through $ID_{V_{a}}=PID_{V_{a_{b,2}}}⊕H\left(sk_{TA}\cdot PID_{V_{a_{b,1}}}\right)$.**Resist various attacks:** Other assaults that our plan can withstand are outlined below.-**Replay attack:** Either in the identity authentication phase or in the message authentication phase, the timestamp tst is included in the messages sent by the vehicle. The repeat of the message could be discovered by SMs and other verifiers by checking the freshness of tst. Therefore, our proposal can resist replay attack effectively.-**impersonation attack:** An attacker must create a valid signature for a message in order to spoof a legitimate vehicle. Based on the above discussion, this is impossible for an attacker and the verifier can easily detect such a malicious attack by validating this signature.-**modification attack:** To alter a message *M* to $M^{\prime }$, an attacker need to generate a valid signature for message $M^{\prime }$. It impossible for the attacker without sender’s private key, and this modified message will be discard by verifiers. As a result, our approach is resistant to modification.-**Stolen Verifier Table Attack:** The proposed sc-heme does not require the verifier to maintain a verification table to complete the authentication. It means that the attacker will not be able to steal any verifier tables for nefarious purposes.

### 5.5. Security Comparisons

We evaluate the security of our proposed approach to three proposed ID-based CPPA schemes [11,14,20] that have been presented.

None of the three previous schemes, according to Table 3, can fulfill all of these security requirements. For Yao [14] and Ali [20], since pseudonym of a vehicle is a constant, they cannot able to realize Unlinkability. In addition, although Lin [11] is able to provide Traceability, TA need to store the relevant public key information for every communication of all vehicles, which is a huge burden for TA. In VANETs, on the other hand, our suggested approach can meet all of these security criteria. On the other hand, TA does not need to maintain any information in order to trace out a vehicle’s genuine identity, which greatly reduces the workload of TA.

## 6. Performance Analysis

In this section, we looked at the performance of the suggested strategy. Firstly, we analyzed the overhead incurred by the proposed scheme, including computing overhead and communication overhead. Then, compared the performance of the proposed scheme with [11,14,20]. Finally, we simulated the scheme based on NS-3 and proved that our scheme is suitable for VANET environment.

### 6.1. Computing Overhead

Due to the pre-set of **Initialization Phase** and **Registration Phase** and **consensus phase** is a completely independent process, we only evaluates the overheads of **Identity Authentication** and **Message Authentication** phases. The following are various notations for execution time.

$T_{bp}$: the time required for a bilinear pairing.$T_{sig-ecc}$: the time required for a signature based on ECC.$T_{ver-ecc}$: the time required for a validation based on ECC.$T_{sm-ecc}$: the time required for a scale multiplication operation based on ECC.$T_{enc-aes}$: the time required for a encryption related to AES.$T_{dec-aes}$: the time required for a decryption related to AES.$T_{pm}$: the time required for a point multiplication operation (i.e., $g_{1}\cdot g_{2}^{-1}$), where $g_{1}$ and $g_{2}$∈$G_{T}$ and the inverse of $g_{2}$ is $g_{2}^{-1}$.$T_{ex}$: the time required for a an exponentiation operation $g_{1}^{\theta }$ where $g_{1}\in G_{1}$ and $\theta \in Z_{\bar{q}}$.$T_{h}$: the time required for a general hash function operation. in *G*.

To compare the computing overhead with [11,14,20] we measured the execution time of the relevant operations through the Java environment with an Intel (R) Core (TM) i7- 8750H CPU 2.20 GHZ and 8 GB RAM. The pairing-based library was used in our simulation and Type A pairings were constructed on the curve $y^{2}=x^{3}+x$ over the field $F_{q}$ for some primes q=3mod4. We performed 1000 times on each operation and ignore the much cheaper operations such as point addition. The average times obtained are shown in Table 4.

To have a better comparison, we separate the communication into two parts to calculate the computing cost, respectively: Vehicle-SM(V2S) communication and Vehicle-Vehicle(V2V) communication. Since TA completes the calculation of public key for all vehicle communication, V2S communication is no longer required for Lin [11]. Instead, we let SMs to do this job to relieve pressure and burden of TA. Similarly, Ali’s [20] scheme does not require this stage because it uses a fixed public-private key pair.

In V2S communication phase, the vehicle can calculate the relevant information in advance based on the public parameters, such as symmetric key, the header and so on. As a result, at this phase, the vehicle just needs to deliver ciphertext to the nearest RSU. Upon receiving the message, RSU is responsible for forwarding it to its region manager SM. When SM receives the message, it will authenticate the message. Similarly, PK is public information, SM only needs to perform two bilinear pairings and one point multiplication operation to calculate the corresponding symmetric key. Then, perform a symmetric decryption operation to get the certificate. Finally, verify the signature by $pk_{root}$ and verify the certificate by using public key of TA. In a word, the execution time of this step is $2T_{bp}+T_{dec-aes}+T_{pm}+2T_{ver-ecc}\approx 22.3196$ ms. For the phase of Yao [14], SM need to execute k+4 point multiplication operations which is tied to the *k* SMs, and perfom three hash operations. Therfore, the execution time is $\left(k+4\right)T_{sm-ecc}+3T_{h}\approx \left(0.7088k+0.0141\right)$ ms.

For the V2V communication phase of our proposed sche-me, vehicle can calculate the subsequent keys and pseudony-ms according to key derivation algorithm in advance. Vehicle sign a message by a new private key and send to nearby RSUs or vehicles. Then RSU will forward this message to SM if this message is a request to access the VFC. Otherwise, adjacent vehicles will verify the message. The verifier(an SM or vehicle) retrieve the corresponding public key by pseudonym and then verfiy the message. Therefore, the execution time of this step is $T_{sig-ecc}+T_{ver-ecc}\approx 11.1461$ ms. On the other side, the execution flow of Yao [14] and Lin [11] are the same as us, so the execution time are also 11.1461 ms. However, for Ali [20], it need to perform two bilinear pairings, three scale multiplication operations, four hash function operations and one exponentiation during this whole process. So the execution time is $2T_{bp}+3T_{sm-ecc}+1T_{ex}+4T_{h}=12.0269$ ms.

To compare the computational overhead of the three sch-emes more clearly, the execution time of four schemes in V2S communication phase and V2V communication phase phases are shown in Table 5. In V2S communication phase, the execution time of our scheme is a constant value. However, in Yao [14], this time will increase with the increasing of *k* (i.e., the number of SMs). Once *k* exceed 32, Yao [14] execution time will beyond ours. In practice, the number of SMs is much bigger than 32. In addition, the execution time during the V2V communication phase is the same for Lin [11], Yao [14] and our scenarios. However, for Ali [20], because of the complicated bilinear pairing process involved in verification phase, the whole communication cost is also the largest among the four shemes. The proposed scheme was reduced about 7% compared with Ali [20]. From the above comparison, our scheme has good efficiency in both stages.

### 6.2. Communication Overhead

We primarily study and compare the communication overhead of our method with three other techniques (i.e., [11,14,20].) in this paragraph. Since **Identity Authentication phase** only needs to be performed once in our scheme, we only consider the communication overhead associated with **Message Authentication phase**. Furthermore, because $\bar{p}$ and *p* have sizes of 64 bytes (512 bits) and 20 bytes (160 bits), the elements in $G_{1}$ and *G* have sizes of 64 × 2 = 128 bytes and 20 × 2 = 40 bytes, respectively.

In each V2V or V2S communication, take the *b*th communication of $V_{a}$ for example, it will broadcast δ={$Sign_{sk_{V_{a,b}}}(M||tst||PID_{V_{a_{b}}}),M,tst,PID_{V_{a_{b}}}$}. $Sign_{sk_{V_{a,b}}}(M||tst||PID_{V_{a_{b}}})$ is a ECDSA signature (i.e., 64 bytes), *M* is a message(where we set as 32 bytes), tst is current timestamp(where we set as 4 bytes) and $PID_{V_{a_{b}}}=\left(PID_{V_{a_{b,1}}},PID_{V_{a_{b,2}}}\right)$. As a consequence, the presented solution is =64+32+4+2×40=180 bytes.

For the same step in Yao [14], $\{Sig_{sk_{v_{b}}}\left(PID_{V_{b}},M,tst\right),$$PID_{V_{b}},M,tst\}$ will be transmitted by $V_{b}$, where $PID_{z}$ is a random pick in $Z_{q}$ (i.e., 20 bytes) that represents its pseudonym. Therefore, the communication overhead is 120 bytes.

In Lin [11], the sender needs to get the transaction identity (TxID, i.e., 32 bytes) from the smart contract by the corresponding $pk_{V_{b}}$ first. Then, $\{Sign_{sk_{V_{b}}}\left(M,tst,TxID\right),M,$tst,TxID} will be transmitted. Here, |pk| is a element of *G* (i.e., 40 bytes). Therefore, the communication cost of Lin [11] is 304 bytes.

Similarly, in the Ali’s scheme [20], the vehicle broadcasts $\left(PID_{a},\kappa_{a},S_{a},tst\right)$ where $PID_{a}=\left(PID_{a,1},PID_{a,2}\right)$, $\kappa_{a}=M⊕h\left(g^{\theta_{a}}\right)$ and $S_{a}=\theta_{a}pk_{v_{a}}$. Since $PID_{a,1},PID_{a,2},S_{a}\in G$, the length of $S_{a}$ is 64 bytes (i.e., suppose the SHA-512 is used), the total communication cost is 188 bytes.

The comparison results are shown in Table 6. It is not difficult to see that the communication overhead of our sche-me is lower than that of Lin [11] and Ali [20]. The communication cost reduces 41% compared with Lin [11] and reduces 4% compared with Ali [20]. Furthermore, even though communication overhead of our scheme is higher than Yao [14], this expense is acceptable because our strategy includes a bonus function(Unlinkability). Therefore, we believe that our solution is more suitable for VANETs in this respect.

### 6.3. Message Authentication Delay and Packet Loss Rate

We also perform a simulation by using NS-3 in a personal computer (Lenovo with Intel Core i7-10875H 2.30 GHz, 16 GB RAM and Ubuntu 20.04 OS) to measure the average message authentication delay and average packet message loss rate. Figure 7 shows a map with 0.5 × 0.5 km^2^ which we adopt in our simulation scenario. The block is maintained by an SM and the interval of broadcast messages is 100 ms. In addition, the packet size in our simulation is 288 bytes. Other parameters like Channel, Propagation, Phy and Mac are set as WirelessChannel, TwoRayGround, WirelessPhy and 802.11, respectively. The simulation time in each simulation are set as 100 s.

Here, we increased the number of vehicles from 5 to 30 and increased the speed of the cars from 7.5 to 40 m/s. The final results are shown in Figure 8. From Figure 8, we can observe that the average packet loss rate is essentially unchanged and until the number of vehicles increases to 30, it increases a little. On the other hand, the average delay increases as the the number of vehicles increases. This could be owing to an increase in the number of vehicles on the road, which would result in an increase in the number of broadcast messages, eventually making a increase in average delay. However, this value is still within our acceptable range.

## 7. Conclusions

With the serious traffic congestion and frequent accidents in real life, VANETs is expected to relieve traffic pressure, for example, changing driving route or reasonable change of traffic lights by the control center through traffic information sent by nearby vehicles. Hence, in this paper, we proposed a hierarchical blockchain-assisted CPPA scheme for VANETs with following advantages: (i) Adopted distributed VFC to reduce the delay effectively in theory. (ii) Attacker unable to link two messages transmitted by same vehicle because the vehicle uses different public key and pseudonym everytime it broadcasts a message. (iii) The proposed scheme has better performance in computing and communication than the previous scheme, which is shown by simulation experiments. The findings reveal that the proposed scheme is available. In the future, how to adopt a more efficient algorithm to design the authentication scheme, whether there is a better way to replace the offline registration of vehicles, will be our focus to improve the efficiency of authentication.

## Figures and Tables

**Figure 1 sensors-22-02299-f001:**
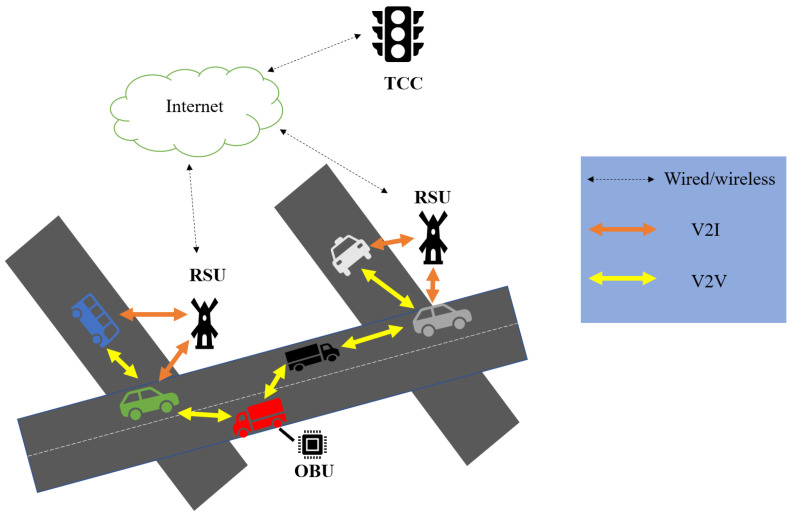
A typical VANET structure.

**Figure 2 sensors-22-02299-f002:**
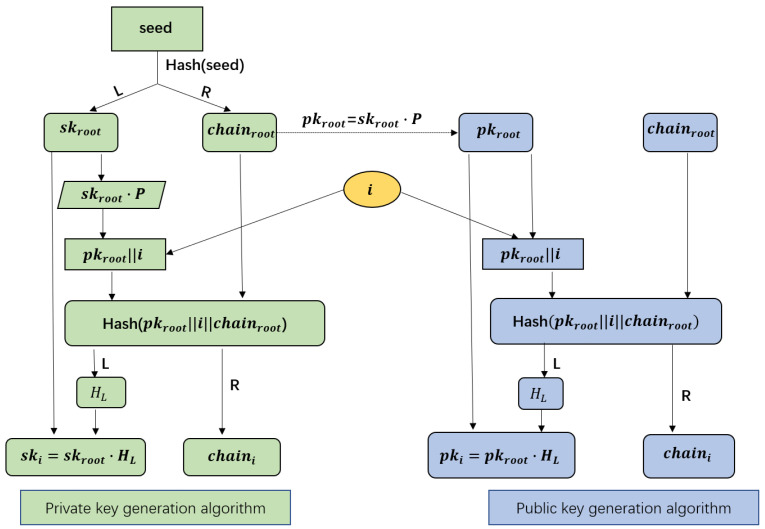
Key generation algorithm.

**Figure 3 sensors-22-02299-f003:**
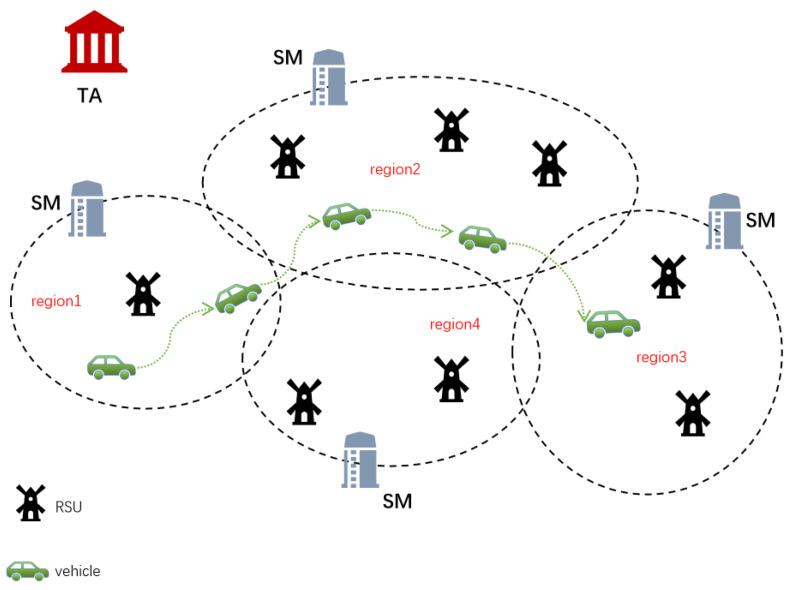
System model.

**Figure 4 sensors-22-02299-f004:**
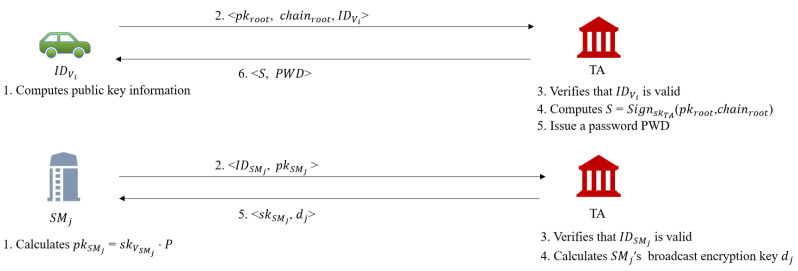
Registration phase.

**Figure 5 sensors-22-02299-f005:**
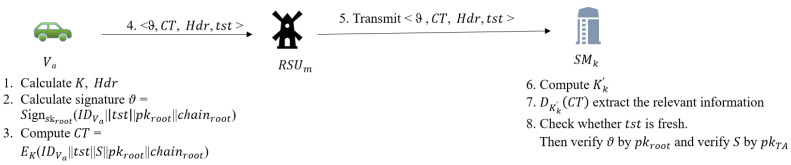
Identity authentication phase.

**Figure 6 sensors-22-02299-f006:**
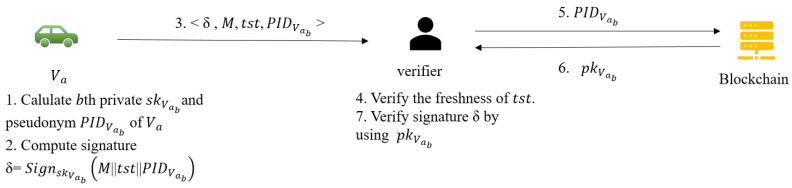
Message authentication phase.

**Figure 7 sensors-22-02299-f007:**
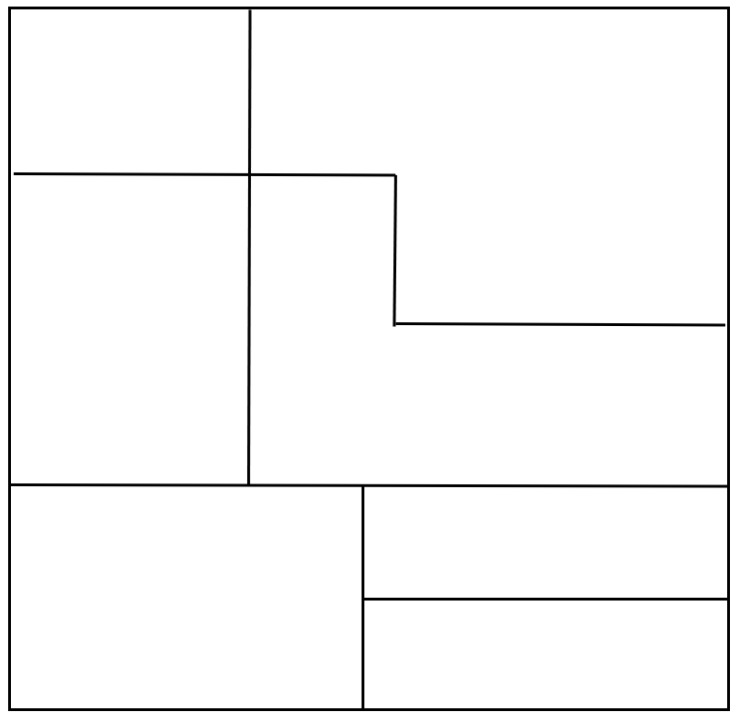
The map with 0.5 × 0.5 km^2^.

**Figure 8 sensors-22-02299-f008:**
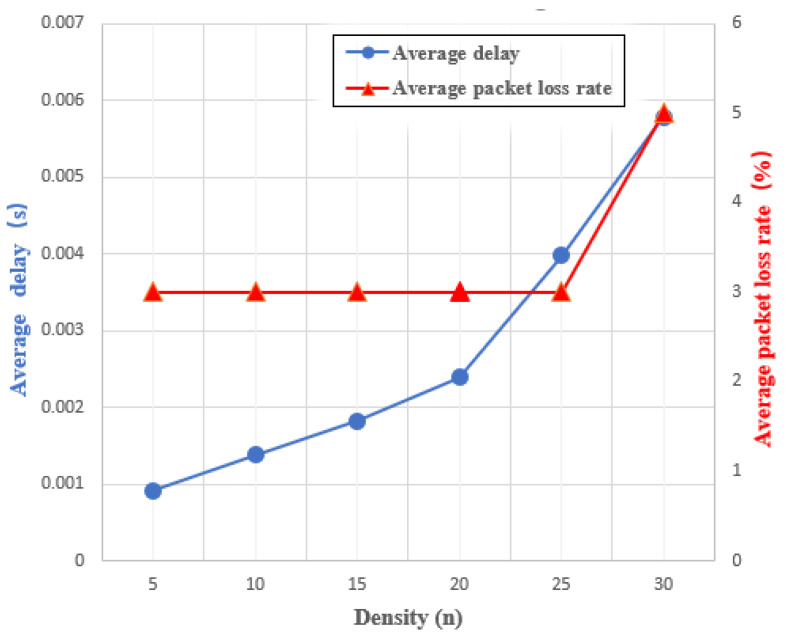
The impact of density in delay and packet loss ratio.

**Table 1 sensors-22-02299-t001:** Summary of related work.

Scheme	Key Technology	Pros	Cons
Picconi [16]	Challenge the aggregator to provide a proof	Easy to verify Independence	Certificate management is difficult
Zhang [17]	*k*-anonymity	Guarantee security and privacy preservation Strong scalability	Certificate management is difficult
He [8]	Elliptic Curve Cryptography	Support batch verification Fast validation	The implementation of cross-region authentication is complex
Zhong [18]	Pseudonym-based signatures	Support batch verification Fast validation	The implementation of cross-region authentication is complex
Wang [19]	Pseudonyms-based and group-based signatures	Easy to verify Guarantee security and privacy preservation	Require frequent authentication
Ali [20]	Hybrid signature	Easy to verify Support batch verification	No consideration for unlinkability
Yao [14]	Distributed VFC Blockchain-based	Flexible cross-region authentication Convenient subsequent certification	No consideration for unlinkability
Kaur [15]	Key-exchange Distributed VFC Blockchain-based	Flexible cross-region authentication Convenient subsequent certification Support mutual authentication	No consideration for unlinkability
Wang [21]	Trustworthiness evaluation mechanism Blockchain-based	Flexible cross-region authentication Guarantee security and privacy preservation	Require frequent interaction
Lu [22]	MPT Blockchain-based	Realize transparency of certificate and revocation Flexible cross-region authentication	Require frequent interaction
Lin [11]	Dynamic key derivative algorithm Smart contract Blockchain-based	Flexible cross-region authentication Guarantee unlinkablility of message	High message processing and queuing latency

**Table 2 sensors-22-02299-t002:** Defintion of notations.

Notations	Definition
*L*	L⊆{1,⋯,n}
$q,\bar{q}$	two large prime integers
*G*	an additive cyclic group of prime order *q*
*P*	a generator of *G*
*n*	the number of SMs
$G_{1},G_{T}$	two multiplicative cyclic groups of prime order $\bar{q}$
*g*	a generator of $G_{1}$
*e*	a bilinear map where $G_{1}\times G_{1}\to G_{T}$
tst	current timestamp
ID	real identity
PID	pseudonym
$E_{K}\left(\right)$	symmetric encryption utilizing *K*
$D_{K}\left(\right)$	symmetric decryption utilizing *K*
*H*	hash function
⊕	exclusive-OR operation
‖	concatenation operation

**Table 3 sensors-22-02299-t003:** Security comparisons.

	Yao [14]	Lin [11]	Ali [20]	Ours
Identity authentication	✓	✓	✓	✓
Message authentication	✓	✓	✓	✓
Identity privacy preservation	✓	✓	✓	✓
Unlinkability	×	✓	×	✓
Traceability	✓	✓	✓	✓
Resist various attacks	✓	✓	✓	✓

✓: The requirement is satisfy. ×: The requirement is not satisfy.

**Table 4 sensors-22-02299-t004:** The average time for each algorithm.

Algorithm	Average Time (ms)
$T_{bp}$	4.6003
$T_{sig-ecc}$	1.5271
$T_{ver-ecc}$	6.5458
$T_{sm-ecc}$	0.7088
$T_{enc-aes}$	0.0434
$T_{dec-aes}$	0.0198
$T_{pm}$	0.0118
$T_{ex}$	0.6871
$T_{h}$	0.0047

**Table 5 sensors-22-02299-t005:** Comparison of computing cost.

	V2S Communication	V2V Communication
Yao’s scheme [14]	$\left(k+4\right)T_{sm-ecc}+3T_{h}\approx \left(0.7088k+0.0141\right)$ ms	$T_{sig-ecc}+T_{ver-ecc}\approx 11.1461$ ms
Lin’s scheme [11]	-	$T_{sig-ecc}+T_{ver-ecc}\approx 11.1461$ ms
Ali’s scheme [20]	-	$2T_{bp}+3T_{sm-ecc}+1T_{ex}+4T_{h}=12.0269$ ms
Our scheme	$2T_{bp}+T_{dec-aes}+T_{pm}+2T_{ver-ecc}\approx 22.3196$ ms	$T_{sig-ecc}+T_{ver-ecc}\approx 11.1461$ ms

-: It does not have this part.

**Table 6 sensors-22-02299-t006:** Comparison of communication cost.

	Communication Cost (bytes)
Yao’s scheme [14]	120
Lin’s scheme [11]	304
Ali’s scheme [20]	188
Our scheme	180

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
