# Peer review of "A Hierarchical Blockchain-Assisted Conditional Privacy-Preserving Authentication Scheme for Vehicular Ad Hoc Networks"

_sensors, 2022, doi:10.3390/s22062299_

Round 1

Reviewer 1 Report

The topic is interesting and consistent with the journal. I have some comments in order to enable the authors to improve their work.

The structure of the paper should be streamlined with a smaller number of sections, maintaining the following contents: introduction; literature review; methodology; results and discussion; conclusions and further research. The literature review should be extended and updated (there are no citations referred to 2020 and 2021).

Add in the abstract a sentence about future research.

I appreciate that the introduction contains information about the research contributions and the description of the structure of the paper. However, a literature review should be addressed more in section 2.

Section 4 and section 5 can be merged.

I find that the paper is developed from a methodological point of view, while the part of discussion and practical implications of the work is a little lacking. Emphasize this in the final part of section 7 and in the conclusions.

The conclusions must be reformulated as they are not consistent at the moment. In addition, they must clearly highlight the ideas for future research.

Reviewer 2 Report

The authors proposed a hierarchical blockchain-assisted conditional privacy-
preserving authentication(CPPA) scheme for VANETs.

I have the following comments to be addressed:

  • The abstract does not have a scenario. is it fog-iov networks? to do what? what is the application?  
  • the performance of the method needs to be clearly outlined with numbers. For example, the computation cost  reduces xx% compared with .... 
  • Figure 2 has been referred 2 pages before. Also, the figure does not reflect the concept clearly.
  • In section 3, looks like to be Preliminaries, not system model
  • It seems the name for sections 4 is system model
  • Section 5 starts with "In this section, we go over our plan in great depth." but it does not reveal any information
  • I would see a table of notations in system model as well.
  • bullet presentations reduces the readability of your paper. Try to get rid of them and explain them into paragraphs
  • There is no connection between the sections. improve the flow of the paper.
  • I would like to suggest the following related papers to be considered:

X. Li, C. Tan, M. Liu, T. H. Luan, L. Gao and Y. Qu, "A Blockchain-Based Cooperative Perception in Internet of Vehicles," 2021 IEEE 94th Vehicular Technology Conference (VTC2021-Fall), 2021, pp. 1-6, doi: 10.1109/VTC2021-Fall52928.2021.9625169.

S. Iranmanesh, F. S. Abkenar, A. Jamalipour and R. Raad, "A Heuristic Distributed Scheme to Detect Falsification of Mobility Patterns in Internet of Vehicles," in IEEE Internet of Things Journal, vol. 9, no. 1, pp. 719-727, 1 Jan.1, 2022, doi: 10.1109/JIOT.2021.3085315.

T. A. Butt, R. Iqbal, K. Salah, M. Aloqaily and Y. Jararweh, "Privacy Management in Social Internet of Vehicles: Review, Challenges and Blockchain Based Solutions," in IEEE Access, vol. 7, pp. 79694-79713, 2019, doi: 10.1109/ACCESS.2019.2922236.

Reviewer 3 Report

The authors dealt with a current issue the paper is well structured, clearly written. - insert representative figures in the introductory part, - - the conclusions must be increased, also referring to future work -the English language is good, check the grammar part, bibliographic suggestions:
Decision tree method to analyze the performance of lane support systems
 G Pappalardo, S Cafiso, A Di Graziano, A Severino Sustainability 13 (2), 846
